# Biofertilizers Improve the Leaf Quality of Hydroponically Grown Baby Spinach (*Spinacia oleracea* L.)

Hayriye Yildiz Dasgan [1,*], Sevda Kacmaz [1], Bekir Bülent Arpaci [1], Boran İkiz [1] and Nazim S. Gruda [2,*]

1  Department of Horticulture, Faculty of Agriculture, University of Cukurova, Adana 01330, Turkey
2  Institute of Plant Sciences and Resource Conservation, Division of Horticultural Sciences, University of Bonn, 53113 Bonn, Germany
*  Correspondence: dasgan@cu.edu.tr (H.Y.D.); ngruda@uni-bonn.de (N.S.G.)

**Abstract:** Plant nutrition through mineral fertilizers is commonly used in soilless culture systems. Our study aims to replace intensive mineral fertilizers with bio-fertilizers, at least partially. We supplemented 50% of the mineral fertilizers with *Chlorella vulgaris* microalgae, a mix of beneficial bacteria and mycorrhiza. In addition, we investigated how to enhance spinach quality by implementing a sustainable and eco-friendly production method. Our research focused on analyzing the parameters of leaf quality and nitrate accumulation of baby spinach grown in a floating culture system utilizing biofertilizers. When mycorrhiza, algae, and bacteria supplemented 50% of mineral fertilizers, 17.5%, 20%, and 21.9% fewer leaf yields than 100% mineral fertilizers (5270 g m$^{-2}$) were achieved. However, biofertilizers improved the internal leaves' quality of hydroponically grown baby spinach. The highest amount of total phenolic (356.88 mg gallic acid 100g$^{-1}$), vitamin C (73.83 mg 100 g$^{-1}$), total soluble solids (9.4%), phosphorus (0.68%), and iron (120.07 ppm) content were obtained by using mycorrhiza. Bacteria induced the lowest nitrate content (206 mg kg$^{-1}$) in spinach leaves, while 100% mineral fertilizers showed the highest nitrate (623 mg kg$^{-1}$) concentration. Moreover, bacteria provided the highest SPAD-chlorophyll (73.72) and titrable acidity (0.31%). The use of microalgae, *Chlorella vulgaris*, induced the highest amount of potassium (9.62%), calcium (1.64%), magnesium (0.58%), zinc (75.21 ppm), and manganese (64.33 mg kg$^{-1}$). In conclusion, our findings demonstrate that the utilization of biofertilizers has the potential to significantly reduce the reliance on mineral fertilizers by up to 50%. Furthermore, an improvement in the quality of baby spinach, as evidenced by an increase in health-beneficial compounds, is possible. Thus, implementing biofertilizers in the cultivation of soilless baby spinach presents a promising approach to achieving both environmental sustainability and improved crop quality.

**Keywords:** plant growth promoting rhizobacteria; mycorrhiza; microalgae; floating culture; soilless culture





## 1. Introduction

World spinach production is 32,294,452 tons, and the amount of spinach produced in Asia and Europe is 30,855,894 and 775,476 tons, respectively. China is the leading producer of spinach, followed by the United States, Turkey, and Japan [1]. In addition to its economic importance, spinach is one of the rising leafy vegetable crops. It is the richest source of minerals providing potassium, calcium, phosphorus, iron, magnesium, manganese, and vitamins K, C, B$_2$, A, and folic acid. Spinach also contains high antioxidants, carotenoids, flavonoids, polyphenols, omega-3 fatty acids, and several health-promoting phytochemicals [2,3].

The popularity of baby leaf vegetables increasingly grows as ready-to-eat leaves with attractive colors, textures, flavors, and rich in health-beneficial bioactive compounds. Baby leaf vegetables grown in hydroponic systems offer consumers convenient and appealing products. These leafy vegetables are harvested at an early maturation stage and prepared

with minimal processing methods and packaging [4]. Decades of research have elaborated detailed strategies for the optimal production of mature leafy vegetables; however, there is a requirement for optimization standards for baby leaf vegetables.

Conventional agricultural practices require high levels of fertilizer to increase crop yields. However, inorganic and chemical-based fertilizers increase production costs, and synthetic fertilizers adversely affect health and the environment [5]. Since the adverse effects of high levels of mineral fertilizers on the environment, bio-fertilizers can be an alternative nutriment or alleviate the hazardous effect of synthetic fertilizers. In addition, biofertilizers can reduce mineral fertilizers, biocontrol pathogens, and increase produce quality [6,7]. The function of biofertilizers could be the fixation of atmospheric nitrogen, synthesis of various phytohormones and enzymes, solubilizing of minerals, and inhibiting phytopathogens [8,9]. Bacteria, mycorrhizal fungi, micro- or macroalgae, and vermicomposts are mainly used for these purposes. For example, it has been determined that bacteria belonging to the genus *Pseudomonas*, *Bacillus*, *Rhizobium*, and *Burkholderia* can make organic phosphate soluble with the help of organic acids they produce and inorganic phosphate with the help of phosphatase enzymes [10]. Furthermore, the bacteria in the plant's root zone support the uptake of nutrients and produce substances such as antibiotics and indole acetic acid, which help plant development [11]. Therefore, bacteria found in the rhizosphere promote plant growth. These beneficial bacteria were named PGPR (plant growth-promoting rhizobacteria).

Non- pathogenic fungi colonizing on root surfaces and living symbiotically with them are named "mycorrhiza". Mycorrhizal fungi dissolve the minerals, including nitrogen, especially phosphorus, through the enzymes and transport them to roots owing to their large hyphae. It is known that mycorrhizae not only support the plant in terms of nutrient uptake but also secretes some enzymes and hormones stimulating plant growth [6,12].

Algae are eukaryotic aquatic organisms that absorb light through photosynthesis and convert inorganic substances into organic substances. They range from small unicellular species (microalgae) to multicellular structures. All algae have photosynthetic mechanisms derived from cyanobacteria and produce oxygen as a byproduct of photosynthesis, unlike photosynthetic bacteria that are not derived from cyanobacteria. Algae can be used as a biofertilizer in living cells or in processed forms. Living microalgae may secrete chelates and allow the plant to benefit from the nutrients better. The microalgae *Chlorella vulgaris* may produce important quantities of plant hormones, polyamines, betaines, auxins, cytokines, gibberellins, and brassinosteroids [13–16].

In modern agricultural aspects, the arrangement of nutrient requirements of the plants is one of the fundamental approaches to reducing the accumulation of nitrate, especially in leafy vegetables [17]. The soilless culture systems allow for arranging inorganic fertilizing of the plants and harvest of clean raw material. Therefore, they can be considered efficient for producing reliable and healthier leafy vegetables [18,19]. In the greenhouse vegetable industry, the focus has traditionally been on yield. However, consumers' interest in recent years in the quality of vegetables, including antioxidants, vitamins, and mineral contents, has increased and will become the driving force in the future [20,21]. Hydroponic cultures significantly contribute to increasing the quality of leafy vegetables by controlling the plants' shoot and root environmental conditions. Furthermore, in recent years, several studies indicated that biofertilizer promotes plant growth and improves product quality [6,7,22,23].

In water culture conditions, biofertilizer studies containing live microorganisms have not been previously encountered in baby spinach growing. To our knowledge, it is the first hydroponic study on spinach under reduced mineral fertilizer conditions with biofertilizers. Soilless growing uses intensive plant nutrition through mineral fertilizers. Our study aims to replace intensive mineral fertilizers with bio-fertilizers, at least partially. Thus, we aimed to increase the product quality of spinach with a sustainable, environmentally friendly production model.

We applied the inoculation of AMF (arbuscular mycorrhiza fungi), microalgae *Chlorella vulgaris* and PGPR to decrease the mineral fertilizers. In addition, we hypothesized that us-

ing biofertilizers, a sustainable, environmentally friendly production model, could increase the product quality of spinach. We investigated baby spinach leaves' nitrate, antioxidants, and mineral contents.

## 2. Materials and Methods

### 2.1. Experimental Design

The study was conducted in a glasshouse during the autumn and winter of 2019 at the University of Cukurova, Adana, Turkiye ($36°59'$ N, $35°18'$ E, 20 m above sea level). Spinach seeds of "Ranchero $F_1$" (Enza Zaden Co., Ltd., Enkhuizen, The Netherlands) material were sown in vials containing mini rock wool cubes on 26 September 2019. The seedlings were transferred to the floating culture system on 26 October 2019. The distance between the rows of basil plants was $15 \times 15$ cm, with a plant density of 44.44 plant m$^{-2}$. Microalgae *Chlorella vulgaris*, AMF, and PGPR were applied to a 50% reduced nutrient solution. We used hard plastic black-colored PVC containers, $105 \times 55$ cm. Plant roots were placed in 50 L containers with an aerated nutrient solution. Each container was used as a replicate according to a randomized blocks experimental design with four replications and 15 plants in each replication.

### 2.2. Plant Nutrition of Spinach

Stock nutrient solutions were prepared as stock A and B (Table 1). Calcium fertilizer is separated from stock A to prevent precipitation of the elements in stock B. Elemental concentrations of nutrient solution are given in (Table 2). Spinach plants were grown in a floating culture system for 90 days and harvested four times by cutting upper leaves (Figure 1). In each harvest, the entire container (15 plants) was harvested. The leaves were cut above the growing tip to enable plants to produce new leaves. The first harvest was made on 23 January 2020, 45 days after transplanting (Figure 2). After that, three more harvests were made at 15-day intervals until 3 March 2020. The nutrient solution was loaded into the containers and kept constant at 50 L. The pH level of the nutrient solution was kept around 6.0–6.2, and EC values were gradually increased at 1.3, 1.8, and 2.0 dS m$^{-1}$ levels in 100% mineral fertilizers control application during plant growth.

**Table 1.** Elemental content and formulations of plant nutrients in stocks used as a reagent.

| STOCK A | STOCK B |
|---|---|
| Calcium nitrate [Ca(NO$_3$)$_2 \bullet$4H$_2$O] | Potassium sulfate (K$_2$SO$_4$) |
| Potassium nitrate (KNO$_3$) | Mono potassium phosphate (KH$_2$PO$_4$) |
| Ammonium nitrate (NH$_4$NO$_3$) | magnesium sulfate (MgSO$_4 \bullet$7H$_2$O) |
| Fe–EDDHA | Micronutrients |
| | Zinc sulfate (ZnSO$_4 \bullet$5H$_2$O) |
| | Boric acid (H$_3$BO$_3$) |
| | Manganese sulfate (MnSO$_4 \bullet$4H$_2$O) |
| | Copper sulfate (CuSO$_4 \bullet$5H$_2$O) |
| | Ammonium molybdate, [(NH$_4$)6Mo$_7$O$_{24} \bullet$4H$_2$O] |

**Table 2.** Concentrations of mineral nutrients in 100% mineral nutrition treatment.

| Element | mg L$^{-1}$ | Element | mg L$^{-1}$ |
|---|---|---|---|
| N | 227.00 | Zn | 0.19 |
| P | 45.00 | B | 0.41 |
| K | 315.00 | Cu | 0.23 |
| Ca | 205.00 | Mo | 0.18 |
| Mg | 75.00 | Mn | 0.78 |
| Fe | 4.62 | | |

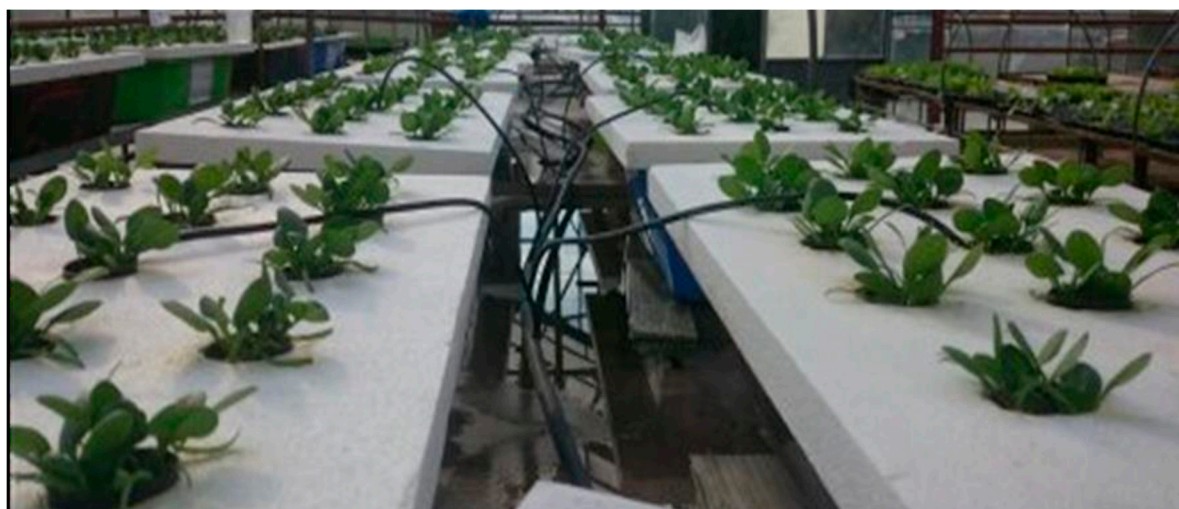

**Figure 1.** The layout of the experiment in the greenhouse.

### 2.3. Biofetilizer Applications

Microalgae biofertilizer containing *Chlorella vulgaris* strain was used. The solution of *Chlorella vulgaris* concentrated $2 \times 10^7$ microalgae mL$^{-1}$ was diluted 40 times as the final concentration and applied to plant roots in float culture [15,16]. Rhizofill™ (NG-Biyoteknoloji Co., Ltd., Istanbul, Turkey), the mixture of *Bacillus subtilis* ($1 \times 10^9$ mL$^{-1}$), *Bacillus megaterium* ($1 \times 10^9$ mL$^{-1}$) and *Pseudomonas fluorescens* ($1 \times 10^{10}$ mL$^{-1}$) was used as the PGPR biofertilizer with a dose of 1.0 mL L$^{-1}$ and supplemented to the nutrient solution [23]. A total of $1 \times 10^4$ g$^{-1}$ ERS™ (Bioglobal Inc. Co., Singapore) mycorrhizal mixture of *Glomus intraradices*, *Glomus aggregatum*, *Glomus mosseae*, *Glomus clarum*, *Glomus monosporus*, *Glomus deserticola*, *Glomus brasilianum*, *Glomus etunicatum*, *Gigaspora margarita* was applied to the seeds before sowing as 1000 spores seed$^{-1}$ [23]. The seedlings were inoculated by mycorrhiza only once during transplanting. With the growth of the root system, mycorrhiza spores multiply in a symbiosis relationship. Meanwhile, to sustain a stable population of microorganisms in the rootzone, regular applications of bacteria and microalgae were applied to the roots at ten-day intervals throughout the growth period [7,23]. The treatments of the study were established as given in Table 3.

**Table 3.** Treatments of mineral and biofertilizer applications of the study.

| Treatments | Explanation |
|---|---|
| 100% Mineral fertilizers (MF) | Without supplementation of any biofertilizer and full of the mineral fertilizers requirements (Control 1) |
| 50% Mineral fertilizers (MF) | Without supplementation of any biofertilizer and half of the mineral fertilizers requirements (Control 2) |
| 50% Mineral fertilizers + Microalgae | Supplemented microalgae to half of the mineral fertilizers requirements |
| 50% Mineral fertilizers + PGPR | Supplemented PGPR to half of the mineral fertilizers requirements |
| 50% Mineral fertilizers + AMF | Supplemented AMF to half of the mineral fertilizers requirements |

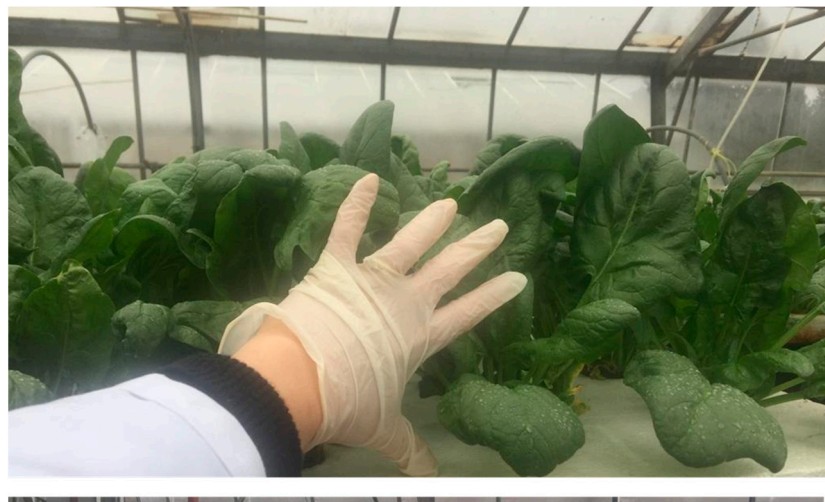

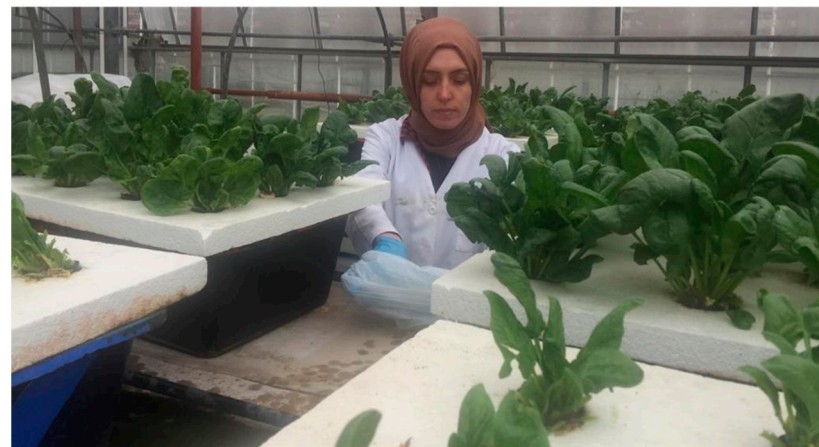

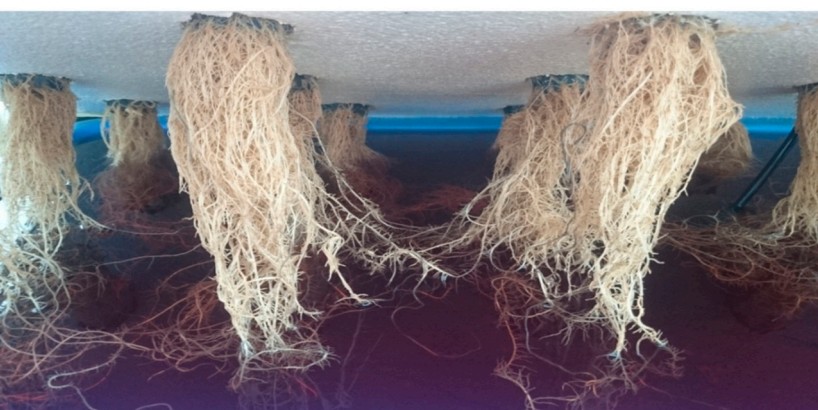

**Figure 2.** Harvest view of baby spinach leaves grown with biofertilizers in a floating culture.

### 2.4. Spinach Leaf Antioxidant Measurements

Antioxidant analyzes were performed on harvested spinach (Figure 3). The ascorbic acid content of spinach leaves was determined using the 2.6 dichlorophenolindophenol titration method. First, spinach leaves were squeezed, and 1 mL of spinach juice was homogenized with 45 mL of oxalic acid (0.4%) and filtered. Then, an aliquot of 1 mL was taken for titration and mixed with 9 mL of 2.6 dichlorophenolindophenol. The absorbance of the samples was measured at 502 nm using UV–vis spectrophotometer (UV-1700 PharmoSpec Shimadzu, Nagoya, Japan) against 9 mL distilled water added in 1 mL sample as the control [23].

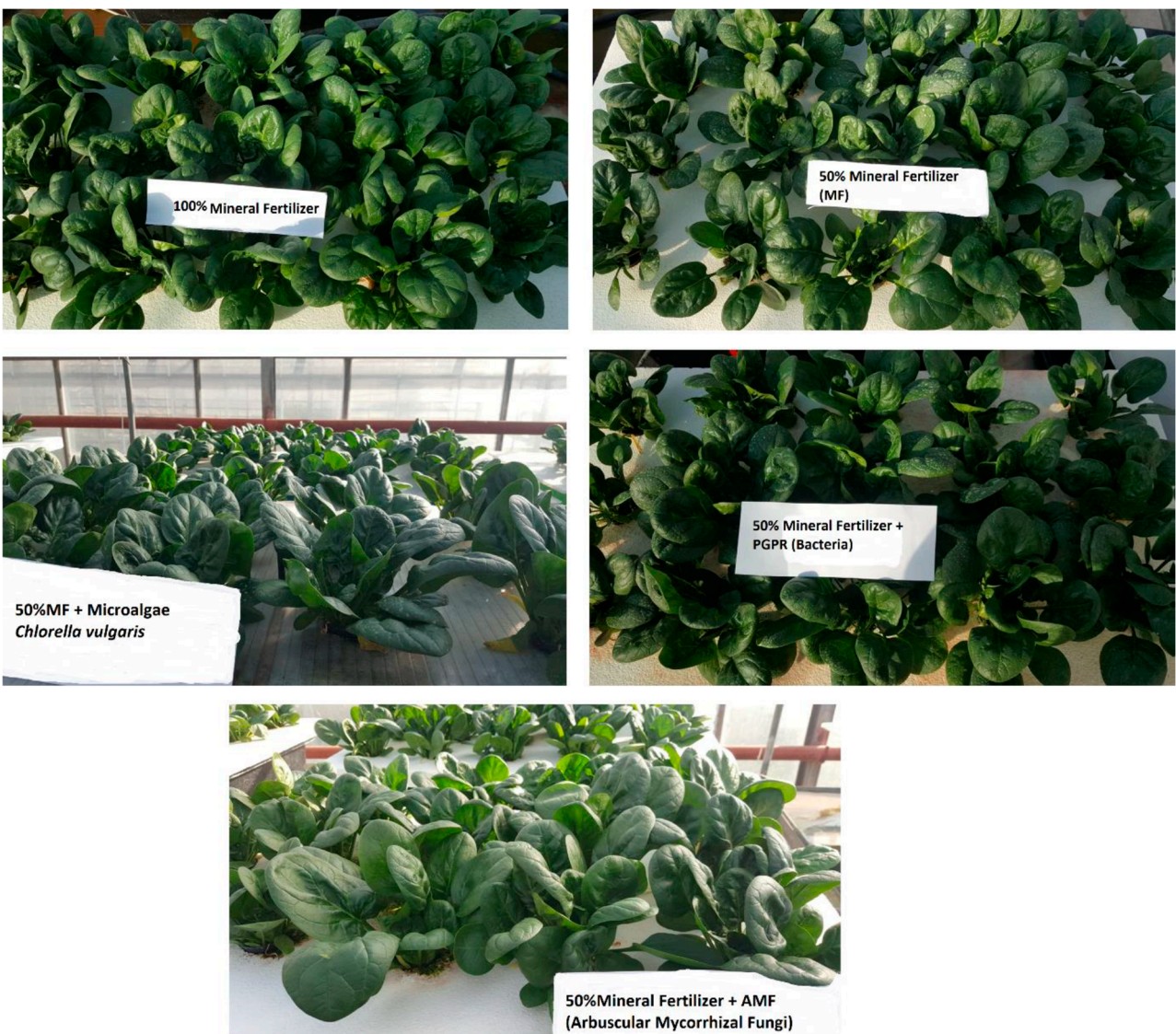

**Figure 3.** Images of baby spinach plants grown hydroponically with supplemented microalgae, PGPR, and AMF in comparison to 100% and 50% mineral fertilizers.

According to Spanos and Wrolstad [24], the total phenolic content was determined with modification. The total extracted phenolic was calculated as milligrams of gallic acid (GA) equivalents using measurements at 765 nm with UV–vis spectrophotometer (UV-1700 Pharma Spec Shimadzu, Japan).

The total flavonoid content of the spinach leaf samples was measured by UV–vis spectrophotometer (UV-1700 Pharma Spec Shimadzu, Japan) at 765 nm, described by Quettier et al., [25]. The total amount of flavonoid substances was calculated by using the calibration prepared with the standards.

Total soluble solids (TSS) were measured with a temperature-compensated digital refractometer (Atago PR-101, Tokyo, Japan) at 20 °C. Titratable acidity (TA) was measured via potentiometric titration (Mettler Toledo DL22, Milton Freewater, OR, USA), and results were expressed in oxalic acid percentage. Finally, a hand EC-meter measured the EC (WTW pH/Cond 3320, Weilheim, Germany). The luminosity (L) and chromaticity [a (red-green axis) and b (blue-yellow axis)] were measured using a colorimeter (CIE Lab, Shanghai, China).

### 2.5. Mineral Elements and Nitrate Analysis

Potassium (K), magnesium (Mg), calcium (Ca), iron (Fe), manganese (Mn), and zinc (Zn) concentrations of spinach leaves were determined by atomic absorption spectrophotometer. Quarters of ten individual plants from each replication were dried at 65 °C for 48 h, and ground using a mill with 20 mesh sieve. The leaf powder was burned in a furnace at 550 °C for 8 h, and the ash was dissolved in 3.3% HCI. Leaf concentrations of Fe, Mn, Zn, and Cu were determined by atomic absorption spectrometry at absorbance mode and K, Ca, and Mg at emission mode [26]. Leaf nitrogen and phosphorus were determined by Kjeldahl and Barton methods, respectively [26]. The colorimetric determination of leaf nitrate ($NO_3$-N) accumulation in the spinach leaves was determined by the transnitration of salicylic acid described by Cataldo et al. [27].

### 2.6. Statistical Analysis

Analysis of variance (ANOVA) was used to test the effect of treatments. Treatment means were compared by least significant difference (LSD), using the statistical software JMP v5.0.1. In addition, all the independent variables were subjected to principal component analysis (PCA) and multiple variable analyses by Pearson correlation matrix ClustVis software (https://biit.cs.ut.ee/clustvis/, accessed on 10 December 2022).

## 3. Results and Discussion

### 3.1. Agronomic Traits

The 100% MF application induced the highest leaf yield with 5270 g m$^{-2}$. Supplementation of PGPR to 50% MF increased the total yield from 3893 g m$^{-2}$ to 4347 g m$^{-2}$. Total leaf yield was 4192 g m$^{-2}$ in microalgae supplementation and 4112 g m$^{-2}$ in AMF (Figure 4). The leaf yield of the biofertilizers remained behind 100% MF in the first two harvests and reached 100% in the third and fourth harvests. Regarding leaves harvested per plant, AMF and PGPR were influential in the first three harvests, while microalgae formed the highest leaf in the last harvest (Figure 4). Although the cumulative leaf weight of four harvests was 87.61 g plant$^{-1}$ in 50% MF, the leaf weight of four harvests were 97.80, 94.33, and 92.53 g plant$^{-1}$ in the AMF, PGPR, and microalgae, respectively. Supplementation of the biofertilizers increased the leaf weight by 12%, 8%, and 6%, respectively, compared to 50% MF.

It has been reported that the macro element concentration of Hoagland solution can be reduced between 10 and 50% in hydroponically grown spinach [28], rocket salad, and parsley [29] cultivations. Öztekin et al. [30] reported 3913 and 3248 g m$^2$ baby spinach leaf yield in a floating culture grown by 100% and 50% mineral fertilizer, respectively. Machado et al. [31] indicated that the leaf yield of spinach grown by coir pith and coir-pith + fiber were 3790 to 4320 g·m$^2$, respectively. Lara et al. [32] reported that the leaf weight of "Viroflay" spinach under different color nettings was between 3.2 and 8.2 g plant$^{-1}$. In hydroponic culture, Levine and Mattson [33] obtained approximately 18 g plant$^{-1}$ baby leaf. The weight of the spinach plant reaches 23.80 g plant$^{-1}$ by organic fertilizers [34]. Dasgan et al. [23] reported that considering the total harvest data, PGPR, AMF, and microalgae treatments increased basil leaf yield compared to 50% MF by about 18.94%, 13.94%, and 5.72%, respectively.

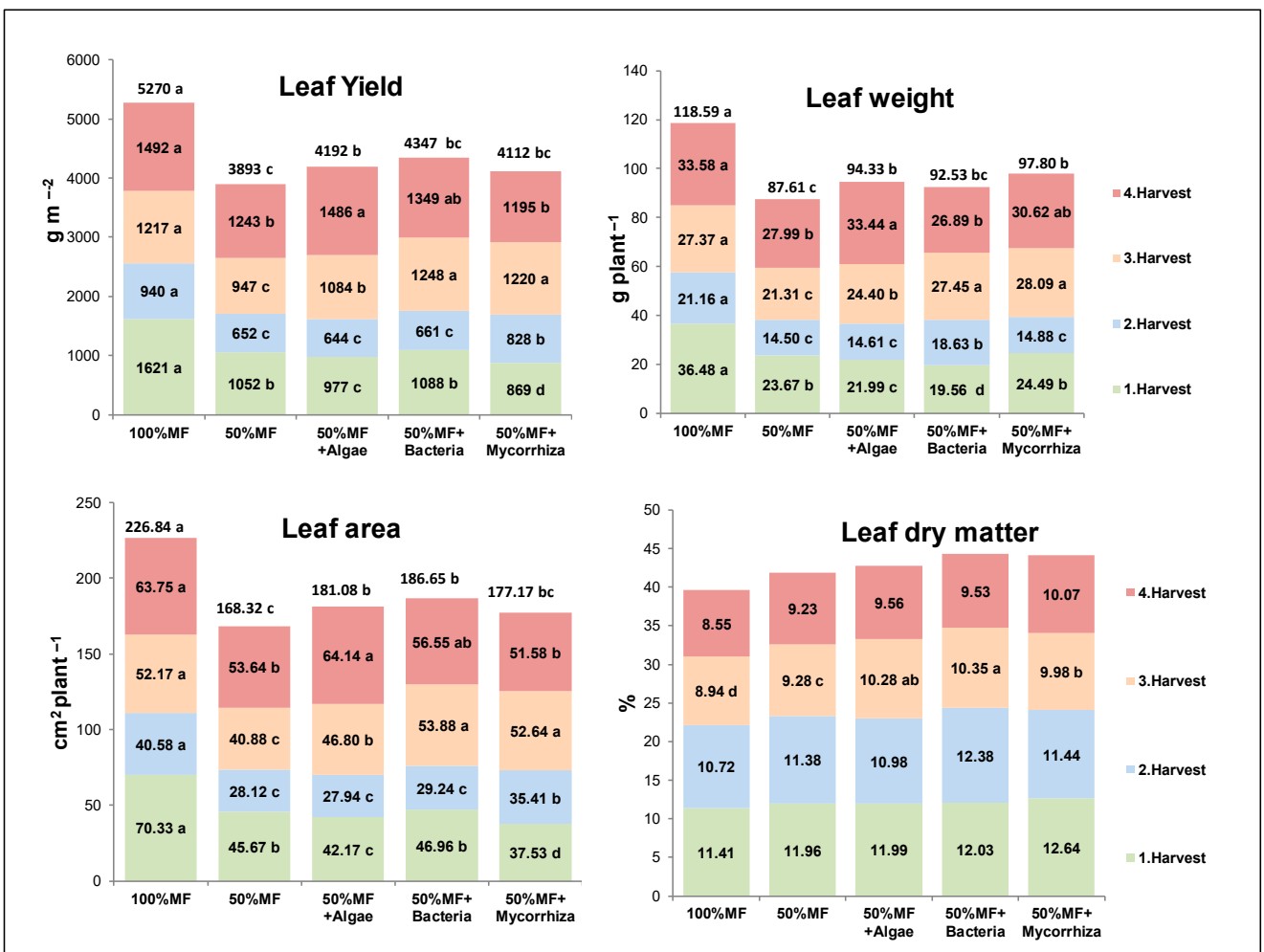

**Figure 4.** Effects of supplemental bio-fertilizers, under the 50% reduced mineral fertilizers, on hydroponically grown spinach baby leaf yield and components. There is no significant difference between means with the same letter in the same color histogram section.

Supplementation of AMF, microalgae, and PGPR to the 50%MF increased the leaf area by 10.88%, 7.75%, and 5.25%, respectively. The highest total leaf area (226.84 cm$^2$ plant$^{-1}$) was obtained in the 100% MF. This was followed by AMF (186.65 cm$^2$ plant$^{-1}$), microalgae (181.08 cm$^2$ plant$^{-1}$), and PGPR (181.08 cm$^2$ plant$^{-1}$) (Figure 4).

PGPR dissolves phosphorus, fixes nitrogen, and increases mineral uptake. They promote the use of nutrients and minerals efficiently and improve shoot growth and root development [35]. PGPR could also improve tolerance to plant diseases and abiotic stresses by altering plant secondary metabolism and detoxifying heavy metals, modulating ethylene levels in plants, and producing volatile organic compounds [36]. AMF also increases the availability and uptake of nutrients. PGPR and AMF increase the plant's photosynthesis by secreting beneficial phytohormones (IAA, cytokinin, and gibberellins), antioxidants, siderophores, enzymes, and vitamins [37]. Jabborova et al. [38] reported that inoculating spinach roots with AMF improved the uptake of nutrients such as N, P, K, Ca and Mg. In addition, the net photosynthetic rate, stomatal conductance, and transpiration rate were increased by the AMF.

Among the treatments, no statistically significant difference was observed in terms of dry matter ratio in the leaf, except for the third harvest (Figure 4). At the third harvest, the highest dry matter ratio in spinach leaves was obtained from the applications with AMF and microalgae, respectively.

### 3.2. Baby Spinach Leaf Antioxidant Properties

While baby spinach leaves with the lowest TSS were obtained from applying 100% MF, reducing mineral fertilizer and additional biofertilizers increased TSS. The highest TSS was obtained from the AMF by 9.4%. The AMF increased the TSS by 10.2% and 16.8% compared to 50% MF and 100% MF, respectively. Dasgan et al. [23] reported an increase in TSS in hydroponically grown basil leaves using AMF. Ergun et al. [15] reported that adding microalgae *Chlorella vulgaris* increased TSS in hydroponically grown lettuce leaves.

A similar response was seen for the total phenol content. The lowest total phenol content (278.87 mg GA 100 $g^{-1}$) was obtained from the 100% MF while reducing mineral fertilizer by 50%, and supplemental biofertilizers increased the phenol content. The highest total phenol was obtained from the AMF by 357 mg GA 100 $g^{-1}$. Avio et al. [22] mentioned that AMF enhanced the biosynthesis of health-promoting phytochemicals, including polyphenols. Howard et al. [39] determined that spinach is rich in phenolic compounds and antioxidants and contains unique phenolic compounds such as patuletins and spinacetins. The authors recorded spinach's phenolic content between 230 and 480 mg 100 g $FW^{-1}$. PGPR and AMF [23] increased the hydroponically grown basil leaf phenolic content.

It was observed that the amount of vitamin C in the spinach leaves increased by supplementing biofertilizers with a nutrient solution in the hydroponic system. Vitamin C values range from 57.27 (50% MF) to 73.83 mg 100 $g^{-1}$ (50% MF + AMF). The vitamin C content of leaves in algae and PGPR supplementation were recorded as 68.11 and 65.41 mg 100 $g^{-1}$, respectively. Addition of AMF, microalgae, and PGPR increased the vitamin C content of the leaves by 28.91%, 18.92%, and 14.21%, respectively. The vitamin C content in spinach has been reported as 19.4 mg 100 $g^{-1}$ [28], 48 mg 100 $g^{-1}$ [40], 25–71 mg 100 $g^{-1}$ [41], and 29.2–58.2 mg 100 $g^{-1}$ [30]. Vitamin C values in this trial were generally higher compared to the literature. It has been reported that vitamin C increased significantly in hydroponically grown basil leaves with AMF and microalgae compared to 50% MF [23.]

Among the applications, the highest acidity (0.31%) was observed in the leaf of 50% MF+PGPR. The lowest acidity (0.21%) was observed in 100% MF. Fusco et al. [42] reported that the inoculation of pepper roots with PGPR increased pepper fruit acidity index, vitamin C, and total soluble solids. Although there was no significant difference between the applications in terms of the EC value of the spinach leaf, the AMF application showed the highest EC. The biofertilizers in this study did not significantly affect spinach leaves' total flavonoid content (Figure 5). However, they ensured the production of flavonoids with as much as 100% MF control. It has been reported that the flavonoid compounds increased significantly in hydroponically grown basil leaves with PGPR compared to 50% MF [23].

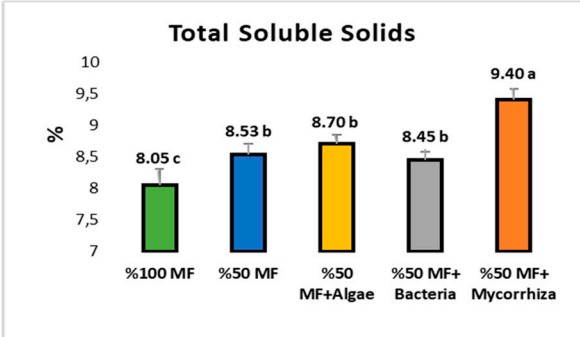
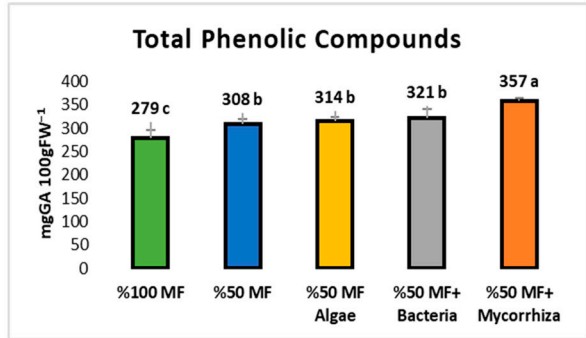
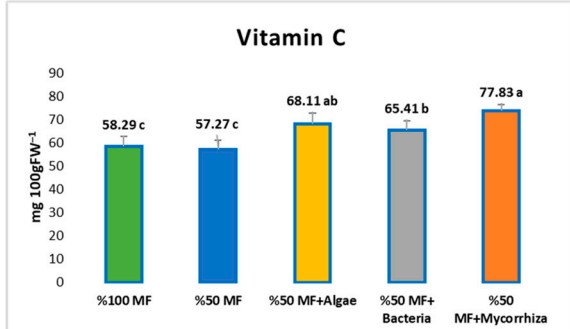
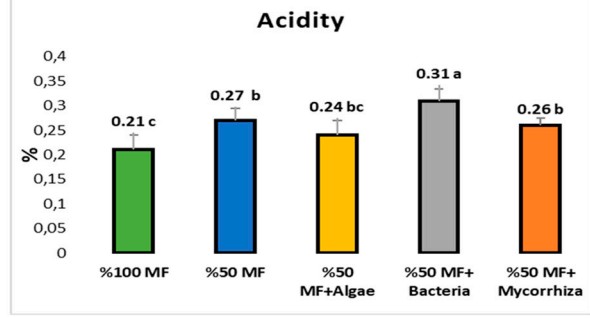
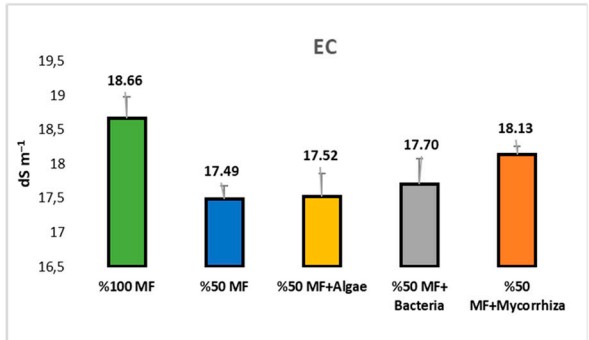
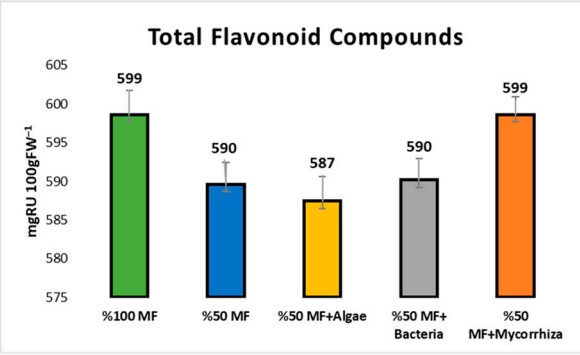

**Figure 5.** Effects of supplemental biofertilizers on leaf TSS (%), total phenolic compounds (mg GA 100 g FW$^{-1}$), vitamin C (mg 100 g FW$^{-1}$), titrable acidity (mg 100 g FW$^{-1}$), EC (dSm$^{-1}$) and total flavonoid compounds (mg RU 100 g FW$^{-1}$). There is no significant difference between means with the same letter in the same histogram, FW fresh weight, GA gallic acid, RU rutin.

### 3.3. Nitrate and Mineral Content of the Baby Spinach Leaf

The absence or the low nitrate content in spinach leaves is essential in terms of being included in baby foods and children's nutritional diets. 100% MF showed the highest nitrate (623 mg kg$^{-1}$ FW) content among all applications. The lowest nitrate concentration (206 mg kg$^{-1}$ FW) was observed in PGPR supplementation, while 386 and 362 mg kg$^{-1}$ FW nitrate were recorded in microalgae and AMF supplementations (Figure 6). The application of PGPR had a significantly reduced effect on the nitrate content. PGPR can alter and regulate enzyme activity in plants. PGPRs may stimulate increases in nitrate reductase enzyme activity. When the nitrate reductase activity increases, as a result, nitrate content in the leaves may decrease over the other treatments [43]. The living *Chlorella vulgaris* cells can cause nitrate reduction in root media and leaves as they use nitrate nitrogen for photosynthesis.

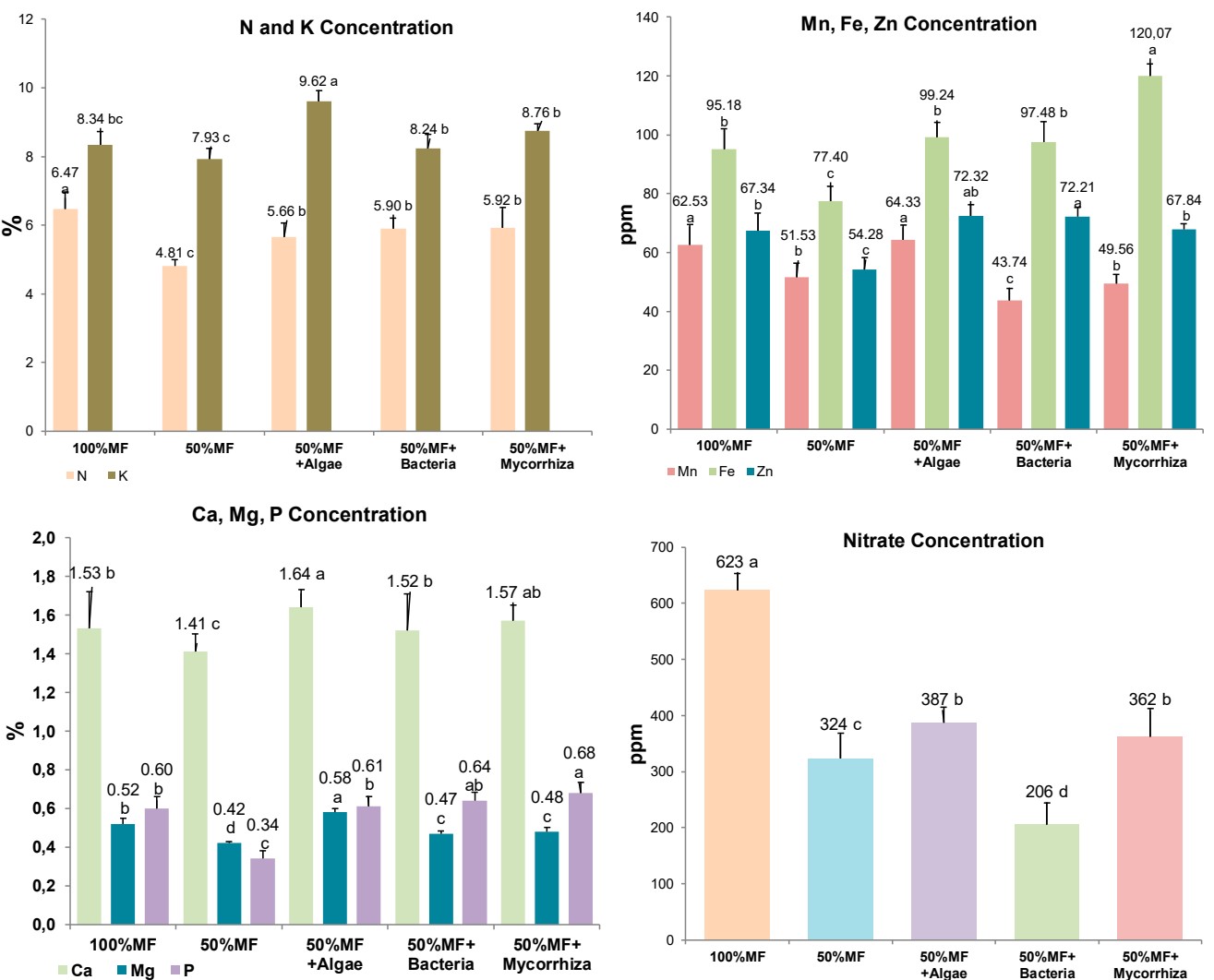

**Figure 6.** Effects of supplemental biofertilizers on baby spinach leaf macro N, P, K, Ca, K (%), micro Fe, Mn, Zn (mg kg$^{-1}$) nutrient concentrations and nitrate (mg nitrate kg$^{-1}$ FW) content. FW: fresh weight. There is no significant difference between means with the same letter in the same color histogram.

Due to their adverse effects on human health, in baby food, nitrate and nitrite are undesirable substances at specific doses. These substances could have a toxic effect causing anemia, and react with secondary amines in the human body to form carcinogenic substances. However, nitrate content in this experiment never exceeded the limits imposed by current EU legislation (3000 mg kg$^{-1}$) [44]. Öztekin et al. [30] reported 1000–1800 mg nitrate kg$^{-1}$ in hydroponically grown baby spinach leaf. Bostanci [45] indicated that the nitrate accumulation in spinach grown in hydroponic culture can be considerably higher than in soil. Dasgan et al. [23] indicated that supplemented microalgae, PGPR, and AMF into 50% MF decreased the nitrate of hydroponically grown basil compared to 100% MF.

Nitrogen (N) content in spinach leaves ranged from 4.8–6.47%. The increase rates were 23%, 18%, and 23% compared to 50% MF in additional AMF, microalgae, and PGPR treatments. Öztekin et al. [30] reported 5.25% nitrogen content in hydroponically grown baby spinach leaf. It is accepted that the plants are adequately fertilized in the case of an abundance of 4% to 6% nitrogen in spinach leaves [46].

The phosphorus (P) content of spinach leaves varied between 0.34 and 0.68%. Through the additional application of microalgae, PGPR, and AMF, the phosphorus content increased by 79%, 88%, and 100% in spinach, respectively. The highest phosphorus content was obtained from the AMF application. AMF provided twice the phosphorus of 50% MF

application (Figure 6). High acquisition of phosphorus by spinach plants could often be mediated by colonizing the root AMF. Plassart et al. [47] mentioned that fungal transport proteins, AMF, possibly express proteins to transfer inorganic phosphate ($P_i$) from the soil or nutrient solution to colonized roots through symbiotic interfaces.

The opposite was regarding the potassium content. The highest potassium content (9.62%) in spinach leaves was obtained from 50% MF + microalgae treatment, whereas Öztekin et al. [30] reported 7.84–8.13% potassium in hydroponically grown baby spinach leaves. Kaynar et al. [48] reported between 2.32% and 13.28% potassium content in spinach leaves. The bio-fertilizers increased the foliar potassium concentration; 50% MF + microalgae, 50% MF + AMF, 50% MF + PGPR applications increased the potassium ratios by 21%, 10%, and 4%, respectively, compared to 50% MF.

The calcium (Ca) concentration of spinach leaves ranged from 1.41–1.64%. The bio-fertilizers increased foliar calcium compared to 50% MF applications. Proportional calcium increases of bio-fertilizer applications compared to 50% MF applications were 16%, 11%, and 8% in the 50% MF + microalgae, 50% MF + AMF, 50% MF + PGPR applications, respectively. Öztekin et al. [30] reported 0.36–0.89% Ca content in hydroponically grown baby spinach. Calcium concentration in the leaves of the spinach plant has been reported in the range of 0.60–1.80% [49].

The biofertilizer also increased the spinach leaves' magnesium (Mg) concentration. Magnesium in the spinach leaf increased by 38%, 12%, and 14% by additional microalgae, PGPR, and AMF treatments, respectively (Figure 6). The Mg concentration range of 0.6% to 1.0% in greenhouse spinach leaves has been reported to be sufficient [50].

Iron (Fe) concentration in spinach leaves varied between 120.07 and 77.40 mg $kg^{-1}$. Fe in spinach leaves increased by 55%, 23%, and 28% by additional AMF, PGPR, and microalgae treatments compared to 50% MF, respectively (Figure 6). The addition of mycorrhiza to the nutrient solution caused a significant increase in leaf iron content. Kaynar et al. [48] reported the iron concentration in spinach leaves in the range of 96–296 mg $kg^{-1}$.

Manganese (Mn) concentration of spinach leaves ranged from 62.53–43.74 mg $kg^{-1}$. The addition of microalgae increased foliar Mn of baby spinach leaves, while PGPR and AMF fertigated leaves were lower than that of 50% MF (Figure 6). Campbell [50] reported sufficient Mn levels in greenhouse spinach leaves in 20–200 mg $kg^{-1}$.

The highest zinc (Zn) concentration was determined in 50% MF + PGPR application (72.32 mg $kg^{-1}$), and the lowest zinc concentration was determined as 50% MF (54.28 mg $kg^{-1}$) (Figure 6). Bio-fertilizers similarly increased the leaf zinc content. With the addition of PGPR, microalgae, and AMF to the 50% MF, the zinc concentration of spinach leaves increased by 33%, 33%, and 25%, respectively. The zinc concentration range of 20–75 mg $kg^{-1}$ in greenhouse spinach leaves has been reported to be sufficient [50].

### 3.4. Color Properties of Spinach Leaves

Luminosity (L) of the leaves was affected by the biofertilizer applications, while other chromatic values, a and b, remained the same. The brightness of the leaves decreased by supplementation of the microbial fertilizer except for algae (Table 4). The highest luminosity was observed in the %100 mineral fertilizers application with 35.38. The AMF and PGPR supplementation decreased the luminosity of the spinach leaves to 28.93 and 24.15, respectively. Moncada et al. [51] observed an increase in the luminosity level of the basil leaves fertilized with organic substance in soilless cultivation, but no significant effect applying plant growth-promoting rhizobacteria (PGPR) was seen. Luminosity is one of the most attractive quality attributes evaluated by consumers to favor foods, especially leafy vegetables. Leaf luminosity is mainly due to its wax content on the leaf surface, which can be affected by the growing conditions. "a" scale indicates; red vs. green where a positive number indicates red and a negative number indicates green. Although there was no statistical difference, the biofertilizers showed more tendency to green color (Table 4). "b" scale indicated yellow vs. blue where a positive number indicates yellow, and a negative number indicates blue. In the present study, the biofertilized spinach leaves

could have a slight tendency to turn green (Table 4). Although there were no significant differences in chlorophyll-SPAD values between treatments, AMF leaves showed slightly higher chlorophyll (Table 4).

**Table 4.** Effects of the biofertilizers on baby spinach leave color's characteristics.

| Treatments | L | a | b | SPAD-Chlorophyll |
|---|---|---|---|---|
| 100% MF | 35.38 ab | −9.0 | 21.4 | 66.33 |
| 50% MF | 37.84 a | −8.3 | 19.6 | 62.15 |
| 50% MF + Algae | 31.58 a–c | −7.4 | 18.0 | 64.64 |
| 50% MF + Bacteria | 24.15 c | −7.6 | 20.6 | 73.72 |
| 50% MF + Mycorrhiza | 28.93 bc | −7.7 | 18.2 | 65.56 |
| LSD$_{0.05}$ | 8.34 | NS | NS | NS. |
| P | 0.03 | 0.08 | 0.71 | 0.1582 |

LSD the least significant difference between the means ($p < 0.05$), NS non-significant. There is no significant difference between means with the same letter in the same column.

### 3.5. Heat Map and Principal Component Analysis

The data of a total of twenty-four parameters; leaf yield, yield components, quality traits, antioxidants, nitrate content and elemental concentrations of spinach baby leaves were calculated by heat map and principal component analysis (PCA) (Figure 7). The highest TSS, total phenols, total flavonoids, vitamin C, phosphorus, iron, and calcium were obtained from 50% MF + AMF. The highest SPAD- chlorophyll, dry matter, b* color value, titratable acidity, and lowest nitrate were obtained from 50% MF + PGPR. The highest leaf potassium, magnesium, manganese, and zinc concentrations were observed from 50% MF + microalgae. The highest leaf yield and yield components, leaf weight, leaf area, nitrogen concentration, nitrate content, and a* color value were obtained from 100% MF, i.e., the control treatment. The highest leaf L* value was observed in 50% MF. Based on our findings, reducing mineral fertilizers by 50% and supplementing AMF, PGPR, and microalgae enhanced the nutritional value and health-beneficial compounds of hydroponically grown baby spinach leaves. Heat maps provide a visual approach to understanding the numerical values. It helps to see many data and their relation to each other in the same picture with color-coded systems. The efficiency of the biofertilizers to plant parameters can be seen all together in Figure 7. The heat maps can also be used in more literal senses, for example, to showcase "hot and cold" (e.g., red to blue) zones on a map. The application of the "50% MF + Mycorrhiza" showed the most red-warm and lightest-blue tones. However, the application of the "50% MF" showed the cold color dark-medium blue tones the most.

The contribution of PC1 and PC2 to variability was 42% and 30.6%, respectively (Figure 7). The principal component analysis (PCA) based on an analysis of twenty-four variables has scattered bio-fertilizers supplementation, mineral fertilizers, and reduced mineral fertilizers in three-quarters of the bi-plot. Both 100% MF and 50% MF treatments disintegrated and dissociated from bio-fertilizer applications. Bio-fertilizer treatment has been grouped and affiliated in the scatterplot.

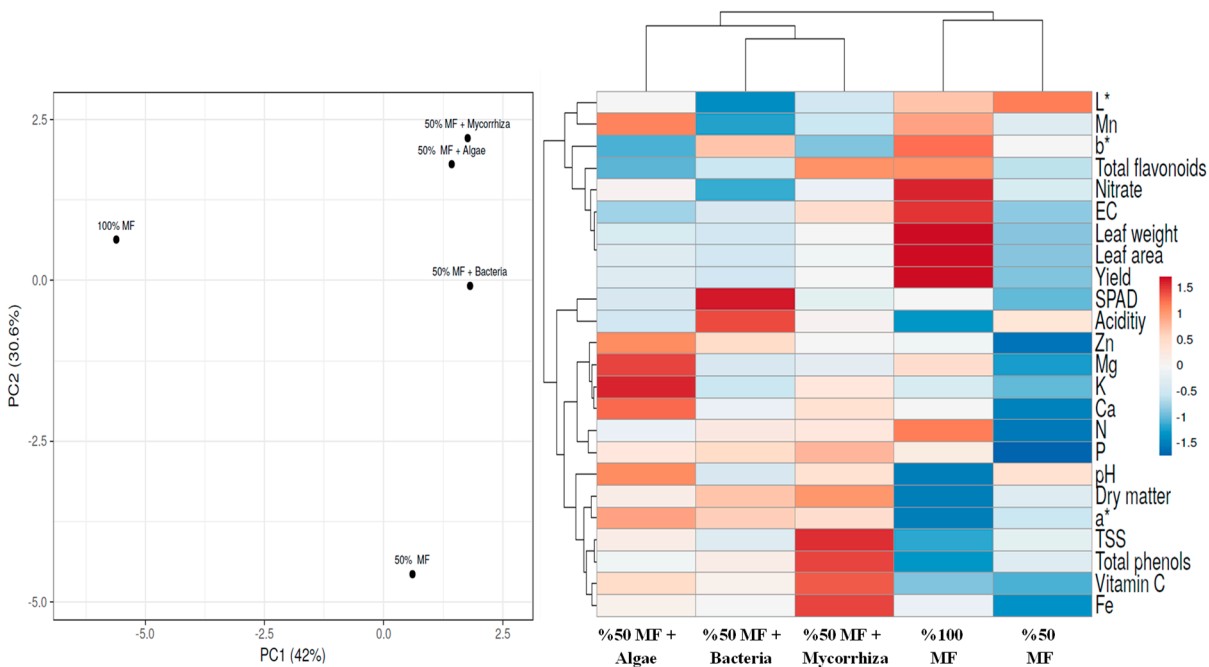

**Figure 7.** Heat map and principal component analysis of bio-fertilizers related to variables.

### 4. Conclusions

Root zone conditions can be easily controlled in hydroponic growing. The present study showed that supplementing AMF, PGPR, and microalgae to the 50% reduced mineral fertilizer in floating culture and alleviates the negative effect of nutrient deficiency in spinach growing. Although the leaf yield and yield components were decreased, quality parameters, total phenols, vitamin C, total soluble solids, chlorophyll, titratable acidity, iron, phosphorus, potassium, magnesium, manganese, and zinc concentrations of spinach leaves were enhanced. In addition, the nitrate content of baby spinach leaf was significantly decreased. In conclusion, our findings demonstrate that the utilization of biofertilizers has the potential to significantly reduce the reliance on mineral fertilizers by up to 50%. Furthermore, an improvement in the quality of baby spinach, as evidenced by an increase in health-beneficial compounds, is possible. Thus, implementing biofertilizers in the cultivation of soilless baby spinach presents a promising approach to achieving both environmental sustainability and improved crop quality.

**Author Contributions:** All the authors contributed to this research. H.Y.D. and S.K. conceived and designed the experiment; data curation; formal analysis; investigation; resources; funding acquisition H.Y.D., S.K., B.İ., supervision; writing—review and editing H.Y.D., B.B.A. and N.S.G. All authors have read and agreed to the published version of the manuscript.

**Funding:** This work was supported by the Cukurova University Research Foundation (BAP) under project number FYL-2019-12445.

**Data Availability Statement:** The data presented in this study are available in the article.

**Acknowledgments:** We thank to Research Foundation Office of the Cukurova University (BAP).

**Conflicts of Interest:** The authors declare no conflict of interest.

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
