# Peer review of "Biofertilizers Improve the Leaf Quality of Hydroponically Grown Baby Spinach (Spinacia oleracea L.)"

_agronomy, doi:10.3390/agronomy13020575_

Round 1
Reviewer 1 Report
The manuscript is interesting and I accepted the revision with pleasure: I think the idea of reducing, even in a floating system, the doses of fertilizer and simultaneously supporting plants with microorganisms opens up prospects for sustainable research. The writing is concise and easy to follow, but the goal of investigating the effects of the treatments 'fertilization reduction' and 'microorganisms' (lines 92-95) should be indicated more clearly. The results are sufficiently explained; all tables and figures have a clear caption, are easily interpreted and not redundant. The findings are supported by the data presented. The manuscript in question is fine and pleasant to read and I do not place major reservations but in my opinion, there is something to review in the experimental methodology.
In detail:
Line 44 requriment - must be corrected
Line 50 biofetlizer - must be corrected
Line 93-95 : you write about a 'control', which? As reported in table 3? %50 Mineral fertilizers? that means without supplementation of any biofertilizer and half of the mineral fertilizers requirements. if it is so and if I have well understood, perhaps this is the most critical aspect of the manuscript. An experiment always includes one control that doesn't receive any experimental treatment. In this case, in my opinion, there had to be a 'control' without fertilizer, with only water and algae-bacteria-fungi, from which you could see the effect of the algae-bacteria-fungi treatment without the 'fertilization' variable. I fully understand that making a spinach floating system with only water (without any fertilizer) wouldn't make sense unless it is miraculous water, but a 'control' with only water and the application of algae-bacteria-fungi can allow observing as they work, their effect without the fertilizer variable, it's a point of comparison against which you can measure test results. But..... I have read reference 23, the experimental design is the same, so I don't beg to differ too much…..Chapeau!
line 140 fig 3 - maybe 2?
line 149: 9ml should be 9 ml (Other cases in the text and captions)
line 148-149 check
line 237 phytochemicals – remove the link
line 248 biotfetilizer - must be corrected
Author Response
Dear Reviewer,
Thank you very much for the nice compliments about our study, and we appreciate your valuable comments and suggestions. We accepted all your suggestions. The changes are highlighted in the manuscript in red color. Please find our answer to your comments below in red colour.
Responses to the Reviewer 1
The writing is concise and easy to follow, but the goal of investigating the effects of the treatments 'fertilization reduction' and 'microorganisms' (lines 92-95) should be indicated more clearly.
Reply
Line 96-99 The following sentences have been added:
Soilless growing uses intensive plant nutrition through mineral fertilizers. Our study aims to replace intensive mineral fertilizers with bio-fertilizers, at least partially. Thus, we aimed to increase the product quality of spinach with a sustainable, environmentally friendly production model.
The manuscript in question is fine and pleasant to read and I do not place major reservations but in my opinion, there is something to review in the experimental methodology.
Reply
Line 107-108 The following sentences have been added:
The study was conducted in a glasshouse during the autumn and winter of 2019 at the University of Cukurova, Adana, Turkiye (36°59′N, 35°18′E, 20 m above sea level).
Line 111-112 The following sentences have been added:
The distance between the rows of basil plants was 15 × 15 cm, with a plant density of 44.44 plant m−2.
Line 147-151 The following sentences have been added:
The seedlings were inoculated by mycorrhiza only once during transplanting. With the growth of the root system, mycorrhiza spores multiply in a symbiosis relationship. Meanwhile, to sustain a stable population of microorganisms in the rootzone, regular applications of bacteria and microalgae were applied to the roots at ten-day intervals throughout the growth period [7, 23].
Line 48 requriment - must be corrected
Reply
Amended Line 48 “requirement”
Line 54 biofetlizer - must be corrected
Reply
Amended Line 54 “Biofertlizers”
Line 93-95 : you write about a 'control', which? As reported in table 3? %50 Mineral fertilizers? that means without supplementation of any biofertilizer and half of the mineral fertilizers requirements. if it is so and if I have well understood, perhaps this is the most critical aspect of the manuscript. An experiment always includes one control that doesn't receive any experimental treatment. In this case, in my opinion, there had to be a 'control' without fertilizer, with only water and algae-bacteria-fungi, from which you could see the effect of the algae-bacteria-fungi treatment without the 'fertilization' variable. I fully understand that making a spinach floating system with only water (without any fertilizer) wouldn't make sense unless it is miraculous water, but a 'control' with only water and the application of algae-bacteria-fungi can allow observing as they work, their effect without the fertilizer variable, it's a point of comparison against which you can measure test results. But..... I have read reference 23, the experimental design is the same, so I don't beg to differ too much…..Chapeau!
Reply
Line 153 We added Control 1 and Control 2 into Table 3 as the reference 23.
line 140 fig 3 - maybe 2?
Reply
Line 158: Corrected as Figure 2
line 149: 9ml should be 9 ml (Other cases in the text and captions)
Reply
Amended. Line 164: 9ml was corrected as 9 ml.
line 148-149 check
Reply
Amended Line 166: “First” was deleted
line 237 phytochemicals – remove the link
Reply
Amented Line 254: We removed the link (underline)
line 248 biotfetilizer - must be corrected
Reply
Amended Line 264 “biofertilizers”

Reviewer 2 Report
The manuscript is, in general, well-written. I have included in the attached file some editorial suggestions and sections that need clarification.
Figure 7
Do you think this figure is necessary? It is not mentioned in the text. What additional information is provided by this figure?

Author Response
Dear Reviewer,
Thank you very much for your valuable comments and suggestions. The changes are highlighted in the manuscript in red color. Please find our answer to your comments below.
Responses to the Reviewer 2
The manuscript is, in general, well-written. I have included in the attached file some editorial suggestions and sections that need clarification.
Figure 7
Do you think this figure is necessary? It is not mentioned in the text. What additional information is provided by this figure?
Reply
We think the heat map and PCA graphic enrich the manuscript and want to save this information. Necessary explanations are given below. Also, additional comments for the heat map graphic have been added to the manuscript (L397-403). Figure 7 is mentioned in the text in several places (Lines 387-400-405).
Figure 7, Line 411: Heat maps provide a visual approach to understanding numerical values. It helps to see many data and their relation to each other in the same picture. Heat maps are graphical representations of data that utilize color-coded systems. They can be applied to data visualizations. Because of their reliance, heat maps are most commonly used to display a more generalized view of numeric values. This is especially true when dealing with large volumes of data, as colors are easier to distinguish and make sense of than raw numbers. However, heat maps are multifaceted and can also be used in more literal senses, for example, to showcase ‘hot and cold’ (e.g, red to blue) zones on a map.
Line 397-403: We add extra comments for the heat map graphic to the manuscript, as follows :
Heat maps provide a visual approach to understanding numerical values. It helps to see many data and their relation to each other in the same picturewith color-coded systems. The efficiency of the biofertilizers to plant parameters can be seen all together in Figure 7. The heat maps can also be used in more literal senses, for example, to showcase ‘hot and cold’ (e.g, red to blue) zones on a map. The application of the 50% MF+ Mycorrhiza showed the most red-warm and lightest blue tones. However, the application of the 50% MF shows the cold color dark and medium blue tones the most.
Corrections in the text have been made as described below:
Line 18 “compound” was deleted.
Line 20“determined” was replaced with “obtained”
Line 24 “with” and “of” were deleted
Line 23 “assumed” was deleted
Line 36-37 corrected as “China is the leading producer of spinach, followed by the United States, Turkey, and Japan”
Line 44 “offering” was replaced with “offer”
Line 53 “used as” was deleted
Line 54 “in” was replaced with “to”
Line 54 “of the” was deleted
Line 54 “produce” was corrected
Line 56 “of” deleted
Line 61 “it is reported that” was deleted
Line 63 “have been observed to” was deleted
Line 70 “intake” was replaced with “uptake”
Line 82“nitrite” was replaced with “nitrate”
Line 109 “hybrid variety” and “used as plant” were deleted
Line 110 September shortened as Sept.
Line 111 October shortened as Oct.
Line 114 sentence corrected: Plant roots were placed in 50 L containers with aerated nutrient solution.
Line 115 “the” was replaced with “a”
Line 118 Did you harvest the entire container or only some plants (how many)?
Reply
We harvested the entire container 15 plants per container
Line 125: In order to explain the following sentence was added to the manuscript
In each harvest, the entire container (15 plants) was harvested.
Line 136 Amended mineral nutrients
Line 184 “In leaves, the” was deleted and the sentence was corrected as Leaf concentrations of Fe….
Line 186 the sentence was corrected “Leaf nitrogen…”
Line 186 “in leaves” was deleted
Line 187 “leaf nitrate” was corrected
Line 191-192: The sentence was corrected as “Treatment means were compared by Least significant difference (LSD)”…
Line 213 the decimal was corrected as 3913 and 3248 g m2
Line 215 was corrected as “Machado et al. [31] indicated that the leaf yield of spinach…”
Reply
Figure 4 (Line 257): In hydroponic production of baby spinach, repeated harvests are applied. Our aim was to show changes in the leaf yield, leaf area and leaf dry matter in 1-2-3-4 harvests, depending on the plant age and the seasonal change in the greenhouse. Therefore, if it is appropriate for you, as the study’s authors, we would like to save Figure 4 as “harvest 1-2-3-4”. Also other referees, did not express a contrary opinion on this issue.
Line 369 corrected as “health-beneficial”

Reviewer 3 Report
I have thoroughly read the manuscript entitled,, Biofertilizers improve the leaf quality of hydroponically grown 2 baby spinach (Spinacia oleracea L.). it is nice work and publishable.

Author Response
Dear Reviewer,
Thank you very much for your valuable comments and suggestions. The changes are highlighted in the revised manuscript in red color. Please find our answer to your comments below in red color.
Responses to the Reviewer 3
Corrections in the text have been made as described below:
Line 9 in the abstract: first write little background , why you conducted this research ?
Reply
Thank you for your feedback. The abstract has been revised to include a brief background. This highlights our focus on enhancing the product quality of spinach through implementing a sustainable and environmentally friendly production model.
Line 13 in the abstract “additional” was deleted
Line 22-24 in the abstract : kindly give the readers, a take home message
Reply
Line 24-29 We change the end of the abstract to:
In conclusion, our findings demonstrate that the utilization of biofertilizers has the potential to significantly reduce the reliance on mineral fertilizers by up to 50%. Furthermore, an improvement in the quality of baby spinach, as evidenced by an increase in health-beneficial compounds, is possible. Thus, implementing biofertilizers in the cultivation of soilless baby spinach presents a promising approach to achieving both environmental sustainability and improved crop quality.
Line 92-96 not your treatment, what was your hypothesis ?
Reply
Line 101-104: A hypothesis is added.
Figure 2 Line 138 i think, first parts of this figure would be deleted, and third root zone part, can be showed with figure 3, using description of this part.
Reply
We kindly want to save Figure 2 as it is. The reason is that the two photos above show the containers used in the study and the hand-measured spinach leaf size. Hand-measured spinach leaf showed the harvest leaf size.
Line 244. Figure 4. kindly show the traits with units on y-axix, and figures legends on the top, not on side ,,and color scheme among harvest can improved like just white , small grid lines etc
i think, in that way, figure will look more appealing ,
Reply
Figure 4 Amended See in the revised MS
Figure 5. Line 260. kindly write units with traits on y-axis, kindly used different color scheme for each treatment
Reply
Figure 5 Amended See in the revised MS
Figure 6. Line 316. kindly write units with traits on y-axis, kindly used different color scheme for each treatment
Reply
Figure 6 Amended, See in the revised MS
Line 394-397: yield decreased but quality increased using biofertilizers particularly Bacteria %50 MF + Mycorrhiza, now what you will give the reader take home message ?
Reply
Amended: Line 423: Implementing biofertilizers in the cultivation of soilless baby spinach presents a promising approach to achieving both environmental sustainability and improved crop quality.
